# Metabolic Alterations in Multiple Myeloma: From Oncogenesis to Proteasome Inhibitor Resistance

**DOI:** 10.3390/cancers15061682

**Published:** 2023-03-09

**Authors:** Philip Weir, David Donaldson, Mary Frances McMullin, Lisa Crawford

**Affiliations:** 1Department of Haematology, Belfast City Hospital, Belfast BT9 7AB, UK; 2Patrick G Johnston Centre for Cancer Research, Queen’s University Belfast, Belfast BT9 7AE, UK; 3Centre for Medical Education, Queen’s University Belfast, Belfast BT9 7BL, UK

**Keywords:** multiple myeloma, cellular metabolism, proteasome inhibitors, drug resistance

## Abstract

**Simple Summary:**

Multiple myeloma (MM) is a common blood cancer that affects plasma cells, a type of immune cell found in bone marrow. Treatment options for MM have improved in recent years, but most patients eventually become resistant to existing therapies, highlighting the need for better treatments. MM cells alter their cellular metabolism to fuel growth and survival and can further adapt their metabolism to promote drug resistance. Here we review the metabolic changes that occur in MM and in the development of resistance to proteasome inhibitors, a common MM therapy, and discuss opportunities for therapeutic intervention.

**Abstract:**

Despite significant improvements in treatment strategies over the past couple of decades, multiple myeloma (MM) remains an incurable disease due to the development of drug resistance. Metabolic reprogramming is a key feature of cancer cells, including MM, and acts to fuel increased proliferation, create a permissive tumour microenvironment, and promote drug resistance. This review presents an overview of the key metabolic adaptations that occur in MM pathogenesis and in the development of resistance to proteasome inhibitors, the backbone of current MM therapy, and considers the potential for therapeutic targeting of key metabolic pathways to improve outcomes.

## 1. Introduction

Multiple myeloma (MM) is a haematological malignancy characterised by clonal plasma cell proliferation in the bone marrow. It is predominantly a cancer affecting older adults. Close to 50% of patients are over 70 years old at diagnosis [1]. Over the past two decades, there have been significant improvements in MM therapy, including the introduction of several generations of proteasome inhibitors (PIs) and immunomodulatory drugs (IMiDs), monoclonal antibodies, the XP01 inhibitor selinexor, and the histone deacetylase inhibitor panobinostat. Despite these advances, and regardless of which drug or combination of drugs is used, MM continues to follow a relapsing remitting pattern in which periods of disease control typically become progressively shorter after each successive line of therapy as the MM cells acquire resistance to treatment. 

The PIs bortezomib, carfilzomib and ixazomib form an integral part of the majority of MM treatment regimens. Whilst bortezomib usage remains the most prevalent, patients now usually receive more than one PI over the course of their MM management. Unfortunately, the development of PI resistance is common during treatment, and in the absence of a cure, new strategies are required to reduce MM resistance to these existing highly effective backbones of therapy and so extend periods of disease control. 

The metabolism of cancer cells is well known to differ from that of their non-cancerous counterparts, and MM is no exception. These adaptations are important to cancer cells—not just to satisfy the demands of increased cell proliferation, but also to modify their cellular processes to better suit the surrounding microenvironment, which is often more hypoxic, acidic and nutritionally deficient [2]. Anti-cancer treatments further alter these cellular processes and the microenvironment. Consequent adjustments to cellular metabolism have often been demonstrated as the neoplasm adapts and mutates to counter the effects of the treatments, gain resistance, and allow continued proliferation.

In this review, we focus on the metabolic adaptations that occur in the development of MM (summarised in Figure 1) and in the subsequent progression to PI-resistant disease, and consider opportunities for therapeutic intervention.

## 2. Proteasome Inhibition

The proteasome plays an important role in cellular protein homeostasis and works as part of the ubiquitin proteasome system (UPS) to regulate protein degradation within cells. Ubiquitin, a small regulatory protein, polymerises and tags unnecessary or damaged proteins for controlled degradation by the proteasome [3]. Degrading these redundant proteins yields peptides which can then be synthesised into new proteins or further degraded to amino acids for re-use. In addition to removing excess unused or misfolded protein, the UPS is vital for numerous normal cellular processes, including growth, differentiation, transcription, and cell cycle control [3].

PIs are very effective at treating MM, as the cells are uniquely dependent on proteasomes to manage the high levels of excess proteins they contain in the form of clonal immunoglobulin. Inhibiting the proteasome in MM therefore stops not just the usual unnecessary and misfolded proteins from being degraded, but additionally blocks degradation of surplus clonal proteins resulting in endoplasmic reticulum (ER) stress. The unfolded protein response (UPR), a series of signalling pathways which regulate ER capacity and homeostasis, is activated to expand the ER and reduce protein synthesis. However, as PIs prevent protein degradation, this process ultimately fails. The proteotoxicity results in further ER stress and ultimately leads to induction of apoptosis of the MM cells [4,5].

Further mechanisms of cell death have been demonstrated after treatment with pIs, including inhibition of the NF-kB pathway. This pro-survival pathway, which has roles in cell proliferation and angiogenesis, relies on proteasomal-mediated degradation of transcription factors for its activation, and this is blocked by pIs [5]. Other proposed apoptotic mechanisms due to pIs include activation of c-Jun NH2-terminal kinase (JNK) and p53, as well as via NOXA, which is a pro-apoptotic BCL-2 family member that activates caspase-9 and further interacts with p53, leading to apoptosis. Specific pro-apoptotic proteins, including Bim, Bid, and Bik, accumulate after proteasome inhibition and are proposed to result in further caspase activation and apoptosis [5]. 

The PIs in current use in treating MM all dually target both forms of the proteasome’s 20S core. The constitutive proteasome encodes proteolytic activities in the β5, β1, and β2 subunits and is ubiquitously expressed. PIs primarily target the β5 subunit. The catalytic subunits β5, β1, and β2 can be replaced with multi-catalytic endopeptidase complex-like 1 (MECL-1) and low molecular mass polypeptide 7 (LMP7) and LMP2 to form the immunoproteasome that is found primarily in haematopoietic cells, with PIs targeting LMP7. Notably, inhibition of both of these proteasome forms is necessary for apoptosis in MM [6]. 

Of the three PIs in general use, bortezomib and ixazomib bind reversibly to their target, whereas carfilzomib binds irreversibly. Bortezomib and ixazomib are boronic acid derivatives, and carfilzomib is epoxyketone-based. Carfilzomib therefore has a different mechanisms of action and is metabolised through peptidase cleavage and epoxide hydrolysis, as opposed to bortezomib and ixazomib, which are metabolised by CYP450 enzymes [6]. These and other general structural and metabolic variations may influence why one PI might still work in MM when resistance has developed to another. It may also explain variations in how MM cells adapt their own metabolic processes in response to the different PIs and their mechanisms of action. In the development of PI-resistant MM, cellular metabolic adaptations that affect any of the above apoptotic pathways could improve cell survival and proliferation and give rise to PI resistance. What is known of these metabolic changes in MM, particularly in response to PIs, is discussed below.

## 3. Carbohydrate Metabolism

### 3.1. The Warburg Effect

Cancer cells are well known to have altered carbohydrate metabolism. The most common example of this is the Warburg effect, where cancer cells favour utilising glucose through aerobic glycolysis with lactic acid fermentation over mitochondrial oxidative phosphorylation, regardless of their oxygen availability. In normoxia, normal human cells mostly metabolise glucose generating pyruvate, which is then decarboxylated and converted to acetyl coenzyme A. This enters the tricarboxylic (TCA) cycle and ultimately undergoes oxidative phosphorylation within the mitochondria, generating up to 36 adenosine triphosphate (ATP) molecules per glucose molecule. Conversely, the aerobic glycolysis of the Warburg effect results in excess lactate and only generates two ATP molecules per glucose molecule [7,8]. 

The Warburg effect with glucose has been extensively studied, along with the corresponding changes in glutamine and fatty acid metabolism, yet some aspects of its function remain incompletely understood [9]. Despite producing more lactate and being significantly less efficient as a means of ATP production than oxidative phosphorylation, aerobic glycolysis is much more rapid and can continue in hypoxic conditions, ensuring ongoing proliferation. Theories for its use include that glucose is rarely in short supply, and so the increased glucose consumed and the excess lactate generated by this pathway may allow for more rapid incorporation of carbon into biomass, maximising the rate of anabolic growth and resulting in faster proliferation, which can potentially outpace the body’s immune responses [9,10,11].

The Warburg effect is known to occur in MM, where despite increased glucose uptake and consumption, oxidative phosphorylation is decreased and lactate production is increased in myeloma cell lines (MMCLs) [12]. This increased glucose uptake is well recognised and can be used diagnostically in patients with MM to identify myelomatous bone lesions on positron emission tomography (PET) scans using the radiotracer fluorodeoxyglucose. 

This aerobic glycolysis of the Warburg effect appears to itself be a potential metabolic target for therapy in MM. The glycolysis-related enzyme pyruvate dehydrogenase kinase-1 (PDK1) is overexpressed in MM but not in normal tissues, and targeting this with the PDK1 inhibitor dichloroacetate was demonstrated to be an effective method of disrupting glycolysis in MM and inducing apoptosis [12].

### 3.2. Hypoxic Microenvironment

Metabolic changes found in the pathogenesis of MM reflect features of both its cellular processes and surrounding bone marrow microenvironment, which is known to have relatively hypoxic conditions. The transcription factor hypoxia inducible factor-1 (HIF1) is a master regulator of the cellular response to hypoxia and has been shown to be upregulated in MM. Increased translation of HIF1A, a subunit of HIF1, is driven by mTOR (mechanistic target of rapamycin) activation via the phosphatidylinositol 3-kinase (PI3K)/Akt signalling pathway. This shifts metabolism by increasing glycolysis and decreasing mitochondrial function through increased expression of glucose transporter 1 (GLUT1), hexokinase 2 (HK2), lactate dehydrogenase A (LDHA), phosphofructokinase (PFK), PDK1, and TCA cycle suppressors [2,7,13], as summarised in Figure 2. Several strategies to target HIF1A, including using HIF1A antisense oligonucleotide EZN-2968, have been investigated in preclinical models of MM and demonstrated reduced proliferation, angiogenesis and bone destruction, along with a metabolic shift to increased oxidative phosphorylation [13,14,15].

HIF1A is also known to contribute to drug resistance in MM cells, and knockdown of HIF1A can overcome resistance to therapies such as bortezomib. LDHA similarly modulates MM drug resistance in areas of hypoxia. Knockdown of LDHA similarly restored the sensitivity of bortezomib-resistant MMCLs [16,17]. Additional support for their importance can be seen with gain-of-function studies of both HIF1A and LDHA inducing resistance in bortezomib-sensitive MMCLs, indicating their potential as therapeutic targets [16].

### 3.3. Glucose Transporters

Gene expression profiling of MM cells showed that they exhibit deregulated expression of GLUT family members, including overexpression of GLUT8, GLUT11, and GLUT4 [18]. MM cells were identified to be dependent on these. GLUT11 and GLUT8 were required for viability and proliferation; and GLUT4 for cell growth, survival, basal glucose consumption, and maintenance of MCL-1 expression. Several metabolism-related drugs have been investigated for their effects on glucose transporters in MM, such as off-target GLUT4 inhibition using the anti-retroviral protease inhibitor ritonavir. Ritonavir, which was already known to modulate the 20S proteasome [19,20], exhibited anti-MM effects as a single agent and was also further studied in combination with the anti-diabetic medication metformin. While this combination demonstrated promising efficacy in pre-clinical models, unacceptable toxicities were reported in a phase 1 trial (NCT02948283) of ritonavir and metformin in patients with relapsed or refractory MM [8,21,22].

GLUT1 has been studied along with HKs, which are important in regulating glycolytic activity. HKs are increased in MM cells, and both they and GLUT1 are downregulated by bortezomib [17]. HK2 knockdown in MM cells results in apoptosis through inhibition of glycolysis and impaired autophagy. HK inhibitors have been investigated as a metabolic target in MM with 3-bromopyruvate and 2-deoxyglucose reducing ATP production and cell viability in MMCLs. Although an enhanced in vitro drug response was observed, unfortunately this was not replicable in vivo [8,23].

The gene serum and glucocorticoid regulated kinase 1 (SGK1), which has roles in cellular glucose uptake, was demonstrated to be highly expressed in the inherently PI-resistant MMCL KMS-20. Activation of SGK1 induced NF-kB p65 phosphorylation, and the high expression in KMS-20 cells was associated with a low sensitivity to ixazomib and bortezomib. Co-treating MM with a PI and either a SGK1 or an NF-kB inhibitor may therefore be synergistic and improve the response to PIs [24].

### 3.4. Oxidative Phosphorylation

Whilst the aerobic glycolysis of the Warburg effect is important in MM, oxidative phosphorylation continues to occur and can be aided by intercellular transfer of mitochondria to MM cells via tumour-derived tunnelling nanotubules from bone marrow stromal cells in their vicinity [25]. This increased mitochondrial access may allow more rapid MM cell proliferation. Given the metabolic consequences of some anti-cancer treatments, such as HK and GLUT1 downregulation by bortezomib, as described above, this may also be a mechanism for MM to alter its metabolism to gain drug resistance, and oxidative phosphorylation is therefore a potential area of metabolism to target to combat resistance.

Both oxidative phosphorylation and the electron transport chain were demonstrated to be downregulated in the PI sensitive MMCL U266-S after bortezomib treatment, through decreases in PGC-1α, cytochrome b, and ATP synthase. These changes were not seen in an isogenic bortezomib resistant line U266-R, indicating that these metabolic proteins can protect the mitochondria against the effects of PIs, and that depleting these may be beneficial in re-sensitising MM cells to PIs [26].

### 3.5. Lactate

Lactate is increased in MM cells through the Warburg effect, as described previously. Lactate dehydrogenase (LDH) catalyses conversions between lactate and pyruvate and is often elevated in active MM. This is important to note, as this metabolic enzyme is recognised in the revised international staging system for MM, which includes raised LDH as an adverse prognostic indicator, since it is associated with increased disease aggressiveness and higher proliferation rates [27].

MM cells transport lactate in and out of themselves via the monocarboxylate transporter (MCT) family. The consequent lactic acidosis of the surrounding microenvironment can result in decreased T-cell proliferation and activation and blunting T-cell function with paresis of both cellular and humoral immunities, highlighting lactate transport as potential therapeutic target [28]. It has previously been shown that MCT1, which is preferentially expressed in MM cells, can be targeted with α-cyano-4-hydroxycinnamic acid to bring about a reduction in the cellular incorporation of lactate and inducing apoptosis in MMCLs [17]. 

### 3.6. Pentose Phosphate Pathway

The pentose phosphate pathway (PPP) is an important branch of glucose metabolism that acts to generate ribose-5-phosphate for nucleic acid synthesis and NADPH, a key component for fatty acid synthesis and antioxidant defence. The PPP has higher activity in bortezomib resistant compared with sensitive MMCLs. This increases the antioxidant capacity of the resistant cells, helping them evade the oxidative stress aspects of bortezomib’s mechanism of action and it appears to relate to higher concentrations of the antioxidant glutathione [29]. Targeting the pentose phosphate pathway, for example, using the G6PD inhibitor 6-aminonicotinamide to reverse the improved antioxidant capacity, would likely have anti-MM effects in bortezomib resistant cells. 

## 4. Protein/Amino Acid Metabolism

### 4.1. Glutamine Metabolism

Glutamine dependence has been identified in MM, with its removal associated with variable degrees of apoptosis [30]. Overall glutaminolysis, the process through which cells covert glutamine into TCA cycle metabolites, is increased in MM cells. Most of the anabolic carbon substrate needed for the TCA cycle is provided through glutamine anaplerosis rather than the usual non-neoplastic method of utilising glucose via pyruvate [31]. The oncometabolite 2-hydroxyglutarate, which can be derived from glutamine anaplerosis into the TCA cycle, was detected at significantly elevated levels in the bone marrow supernatants of patients with MM than in those with the precursor condition monoclonal gammopathy of undetermined significance (MGUS) and correlates with higher levels of MYC expression [31]. It has been shown in B-cells that the increased contribution of carbon to the TCA cycle over glucose is particularly seen under hypoxic conditions and is also associated with MYC [32]. 

MYC expression is not typically altered in MGUS but is upregulated in around 70% of patients newly diagnosed with MM and is believed to drive MM cell dependence on glutamine metabolism through mechanisms promoting the transcription of the cell membrane glutamine importers ASCT2 and SN2 [31]. This relationship between MYC protein expression and glutamine metabolism in MM cells was further elucidated with inhibition of key glutaminolysis enzyme glutaminase using compound-968, inducing apoptosis via MYC degradation [33]. 

Another mechanism explored to target glutamine addiction in MM cells is using L-asparaginase, an anti-cancer enzyme used to treat acute lymphoblastic leukaemia. Treatment with L-asparaginase depletes asparagine and glutamine levels through hydrolysis, leading to anti-MM activity as a single agent and a synergistic effect in combination with PIs [8].

Inhibiting glutamine uptake in MM cells via ASCT2 using the amino acid analogue GPNA resulted in markedly decreased cell growth. In addition to the cell-membrane glutamine importers ASCT2 and SN2, the glutamine transporters LAT1 and SNAT1 are also more highly expressed in MM cells [34]. The consequent increased glutamine uptake from greater transporter expression has been noted to relate to significantly higher amounts of glutamine being taken up by bone marrow MM cells than by the remainder of the bone marrow mononuclear cells [31]. 

The importance of glutamine has been further demonstrated in PI resistance: MM cells resistant to bortezomib and carfilzomib switched their metabolism to increasingly utilise glutamine in conditions where glucose metabolism was inhibited [17]. Enhanced mitochondrial function fuelled by glutamine instead of glucose was observed in bortezomib-resistant MMCLs, reinforcing the importance of glutamine and related pathways as therapeutic targets for resistant MM [2]. Some glutamine related targets are under investigation in this context, such as glutaminase inhibitor CB-839, which worked synergistically with several PIs at enhancing their cytotoxic activity against both sensitive and resistant MMCLs [35] and is currently in a phase 1 trial in combination with carfilzomib and dexamethasone for relapsed or refractory MM (NCT037986780). Glutamine transporters have also been targeted with ASCT2 inhibition working synergistically with PIs to increase cytotoxicity in PI-sensitive and -resistant MMCLs [36]. 

In addition to inhibition by lactic acidosis, T-cells and NK-cells appear to be suppressed by hypoxia and nutrient deprivation in the bone marrow microenvironment through MM’s increased glutamine dependence [17]. This may further help MM cells proliferate by avoiding immune regulation. Targeting this area of metabolism has the potential to inhibit MM cells both directly and indirectly through restoration of T-cell function.

Overall, glutamine dependence appears to make a significant contribution to MM progression and the development of drug resistance, and this remains a relatively unexplored area that displays a number of opportunities for therapeutic targeting.

### 4.2. Serine Synthesis

Serine plays an essential role in MM proliferation by providing precursors for the synthesis of proteins and nucleic acids. Bortezomib resistant MMCLs were shown to have both increased serine uptake and increased serine synthesis. Serine starvation reduced cell viability and was also seen to enhance the cytotoxic effects of bortezomib in PI-sensitive MMCLs, indicating the importance of serine to the cells’ ability to overcome bortezomib treatment [29]. Partial PI resistance in MM was demonstrated to be caused by mitochondrial proteases, and overexpression of the serine peptidase lon peptidase 1 (LONP1) compensated for the reduced proteasome activity [37].

An important enzyme in the serine synthesis pathway, phosphoglycerate dehydrogenase, was targeted with the small molecule inhibitor NCT-503. This reduced the intracellular redox capacity of the cells and increased sensitivity to PIs in resistant MMCLs, and exhibited therapeutic advantages in vivo [38]. This pathway may therefore have great potential for therapies targeting metabolism, and further study of other pathway targets may identify additional treatments.

### 4.3. Other Amino Acids

Whilst glutamine appears to be the most important amino acid altered in MM metabolism, many other amino acids also exhibit variability. Isoleucine and threonine are both significantly decreased in filtered plasma obtained from bone marrow samples of patients with both MM and MGUS compared with healthy donors. Another study using serum samples of patients with newly diagnosed MM found that the levels of the amino acids isoleucine, arginine, phenylalanine, and tyrosine were elevated; and those of lysine and glutamine were decreased, compared to samples from healthy controls, indicating changes involving protein expression or metabolism. The same study identified increased amounts of acetate and decreased 3-hydroxybutyrate, and some lipid variability [17].

A metabolomic study profiling peripheral blood plasma samples from patients with newly diagnosed MM against those with relapsed/refractory disease identified 13 significantly altered metabolites, including free carnitine, acetylcarnitine, and creatinine [39]. The same study identified 36 amino acids and biogenic amines that had significantly different concentrations between control and MGUS patient samples, as well as 8 metabolites, including asymmetric dimethylarginine, free carnitine, acetylcarnitine, and glutamate, that were significantly altered between MGUS and newly diagnosed patient samples. This study demonstrated how variations occur in metabolism throughout the cancer progression process, from normal controls, to MGUS, to newly diagnosed and then relapsed/refractory MM. The relevance of this is unclear, however, as these changes were only those significant enough to be found in the peripheral blood, where MM cells do not usually circulate. More numerous or alternative significant changes may be expected if studying metabolic differences of MM cells or their bone marrow microenvironment.

Three important amino acids, valine, leucine, and isoleucine, were noted to be at lower concentrations in newly diagnosed MM. These branched-chain amino acids found exclusively in the diet have been previously linked to cancer progression and are used as signalling molecules for energy production and in protein synthesis [39]. Since they cannot be manufactured by the body, their lower concentration in MM suggests an increase in their utilisation by the cancer cells, and targeting them may therefore have therapeutic benefits.

Both glutamate and urenine, a metabolite of the essential amino acid L-tryptophan, were elevated in new diagnosed MM patients. Glutamate has vital roles in oncogenic signalling, in disposal of excess nitrogen, and as a key substrate for cellular metabolism. Kynurenine (KYN) has a role in numerous cellular functions with related enzymes having been shown to be involved in the development and progression of cancer through immune suppression, immune tolerance, and their involvement in inflammatory processes in the microenvironment [39]. Targeting IDO1, the rate limiting enzyme in the KYN pathway has emerged as a potential anti-cancer therapy in recent years. Early clinical trials in solid tumours produced disappointing results; however, this remains an active area of immune-oncology results [40]. With higher concentrations and such important functions, targeting aspects of glutamate or kynurenine metabolism may be a promising approach in treating MM.

Glutamine-to-proline conversion is aided by mitochondrial enzymes, including pyrroline-5-carboxylate reductase 1 and 2 (PYCR1 and PYCR2). Overexpression of PYCR1 and PYCR2 mRNA was shown to be associated with worse overall survival in patients with MM, and in vitro studies identified that pargyline, a selective monoamine oxidase B inhibitor that inhibits PYCR1, worked synergistically with bortezomib in inducing apoptosis in the MMCL RPMI-8226 [41].

Glutathione, generated from the amino acids cysteine, glycine, and glutamic acid, is a major intracellular antioxidant with numerous metabolic roles. Although reactive oxygen species are generally elevated in cancer, including MM, the cells continue to undergo apoptosis if these reach toxic levels. Adequate levels of glutathione are therefore necessary for cell survival, and glutathione degradation was demonstrated to be a potential target in overcoming bortezomib-resistant MM through altering intracellular oxidative balance. Pre-treatment of the MM cells with either of the omega-3 fatty acids docosahexaenoic acid or eicosapentaenoic acid leading to increased apoptosis [42].

## 5. Fatty Acid/Lipid Metabolism

Obesity is known to increase the risks of developing MM and of MM associated mortality. It is associated with a generalised increase in adipocyte numbers, leading to increased inflammation, accumulation of fatty acids, and dysregulated secretion of adipokines. While subcutaneous and visceral adipose tissue largely account for the increased adipocyte numbers, obesity also induces adipocyte accumulation in the bone marrow. Furthermore, as patients age, their bone marrow contains notably increased quantities of adipocytes relative to haematopoietic cells, which may be relevant to the increased incidence of MM in older people. MMCLs cultured with adipocyte-conditioned media have been demonstrated to proliferate more rapidly, while having increased cell adhesion and increased pSTAT-3/STAT-3 signalling. When the adipocytes were taken from obese or “super-obese” patients, they displayed increased PPAR-γ, cytochrome C, and pro-tumourigenic adipokines IL-6, and leptin, along with reduced levels of the tumour suppressive adipokine adiponectin compared with adipocytes taken from individuals with a normal BMI [43]. Bone marrow adipocytes additionally appear to have an effect on the phenotype of MM cells, e.g., through the supply of additional free fatty acids to support growth and other functions [44]. Therapeutic benefit may therefore be obtained by targeting this metabolic relationship.

Obesity may also increase risks of MM via other commonly linked metabolic syndrome factors, such as hypercholesterolaemia, hyperlipidaemia, hypertension, and diabetes. These metabolic syndrome conditions typically involve increased levels of oxidatively modified low-density lipoprotein (OxLDL), which has been associated with stimulation of pro-oncogenic and survival signalling and an increased risk of developing solid tumours. OxLDL was demonstrated to be present in the bone marrow microenvironment of MM, and was further shown to exhibit cytoprotective effects in both primary MM cells and MMCLs against the boronic acid-based PIs bortezomib and ixazomib, indicating that it is a mediator of chemoresistance. Interestingly, these effects were not significant against the epoxyketone-based PI carfilzomib, which continued to work effectively and provides an example of how different classes of PIs are resisted through different metabolic pathways [45]. Statin therapy is known to reduce OxLDL levels, and a study of almost 5000 patients assessing the effects of statin use following or shortly before a diagnosis of MM was associated with MM-specific mortality being reduced by 24%, and skeletal-related events were decreased by 31%, suggesting a role for these agents in MM management [46]. Elsewhere, the statin simvastatin has been seen to exhibit anti-MM activity in vitro [47], and a small phase II trial (NCT00399867) demonstrated a synergistic effect of simvastatin in combination with PI treatment, even in PI refractory patients [48].

Saturated fatty acids and n-6 polyunsaturated fatty acids were shown to be increased in the erythrocyte membranes of patients with MM compared to controls. Monounsaturated, n-3 polyunsaturated, and trans-fatty acids were reduced [49]. Another analysis of lipid metabolism showed that MM cells had significantly decreased levels of the phospholipid phosphatidylcholine compared with normal plasma cells [50]. Elsewhere in MMCLs, in conditions of decreased glycolysis and increased lactate accumulation, it was found that metabolic homeostasis could be maintained through increased fatty acid oxidation, which again points towards methods of metabolic switching that may be possible to target [17].

The expression of fatty acid synthase, the enzyme responsible for biosynthesis of fatty acids, is upregulated in both primary MM cells and MMCLs. Inhibition of fatty acid synthesis using orlistat significantly reduced MM cell proliferation, and it has consequently been suggested that fatty acid metabolism may be an effective therapeutic target for MM. This was further supported through another mechanism with inhibition of fatty acid oxidation by etomoxir also showing reduced proliferation [17].

A metabolomic study also demonstrated differences in fatty acid plasma concentrations: eight lysophosphatidylcholines, a class of inflammatory lipids, were found at lower concentrations in newly diagnosed MM patients compared to healthy controls. Some lysophosphatidylcholines and phosphatidylcholines concentrations were altered further in the progression to relapsed/refractory MM [40].

### Lipid Metabolism and PI Resistance

Overall understanding of the changes in lipid metabolism of resistant MM cells is poor, but increased lipogenesis has been identified. Treatment of MMCLs with eicosapentaenoic acid or docosahexaenoic acid, both polyunsaturated fatty acids, resulted in apoptosis and decreased proliferation, which appears to work via lipid peroxidation. The addition of dexamethasone to these drugs had a synergistic effect, even in the cell line MM1R, which is known to be dexamethasone resistant, and suggests their use may be a possible way of overcoming resistance [17]. Fatty acid elongase 6 levels were noted to be lower in bortezomib-resistant MM, and its restoration resensitised the cells to bortezomib, indicting a further potential treatment strategy [51].

Proteasome inhibition results in significant accumulation of lipids in MMCLs, and abnormal lipid metabolism is induced via the ATF4/SREBP pathway. The expression of ATF4, which has roles in lipid metabolism and the UPR, was significantly increased following proteasome inhibition. Combining treatment of a PI and either the lipid-lowering drug lovastatin or a fenofibrate derivative proved synergistic in inducing cytotoxicity in MMCLs [52].

PI-resistant MMCLs AMO-BTZ and AMO-CFZ were found to accumulate mono-acylglycerols and undergo lipid class switching from lysolipids to sphingomyelins compared to PI-sensitive cells. The increased synthesis of sphingolipids was exploitable using a sphingomyelin synthase inhibitor D609, which was associated with increased cytotoxicity in the PI-resistant MMCLS [53]. Bortezomib- and carfilzomib-resistant MM was also shown to be resensitised to either bortezomib or carfilzomib using the sphingosine kinase 2 inhibitor K145. This appeared to work via enhanced activation of the UPR and indicates that targeting altered sphingolipid metabolism may be worth further exploration in the management of PI-resistant MM [54].

Coenzyme Q10, a mitochondrial electron carrier and product of the cholesterol producing mevalonate pathway, was shown to be relevant in bortezomib resistant MM. Bortezomib resistant MM appears to have elevated vulnerability to electron transport chain inhibition and targeting the biosynthesis of coenzyme Q10 had a synergistic effect with bortezomib in treating bortezomib resistant MM [55].

## 6. Nicotinamide Metabolic Pathways

In MMCLs resistant to bortezomib and carfilzomib, quantitative and functional proteomics identified significant alterations in energy and redox metabolism. Protein folding and destruction, metabolic regulation, and redox homeostasis were upregulated. Metabolic changes favoured elevated levels of reducing agents such as nicotinamide-adenine–dinucleotide–phosphate (NADPH), which protects cells against oxidative damage. It was theorised that the increased generation of NADPH and the associated increased antioxidant capacity could make MM cells more tolerant of proteasome inhibition and that manipulation of the energy and redox metabolism may therefore be good targets for overcoming PI-resistant MM [2,56].

These findings were replicated in a bortezomib-resistant U266-R MMCL compared to the PI-sensitive U266-S MMCL, where NADPH was seen at higher concentrations in the resistant cells. Furthermore, whilst concentrations of both reduced and oxidised nicotinic coenzymes such as NAD^+^ were similar between U266-S and U266-R under basal conditions, U266-S alone showed significant depletion of these after treatment with bortezomib [25]. This suggests that alterations to NAD^+^ metabolism are significant in bortezomib resistance and further exploration in this area may derive therapeutic benefits in resistant MM.

## 7. Gene Expression Signatures

Many of the metabolic variations that occur in the oncogenesis of MM and in its subsequent acquisition of PI resistance can be identified through examining the altered genetics of the disease. Altered expression of some genes known to regulate MM metabolism were shown to correlate with 5-year survival. Seven of these are used to generate a metabolic risk score. *CISH* under-expression was associated with favourable survival; and *NSDHL*, *CTPS1*, *FABP5*, *SLC25A5*, *FLNA*, and *UBE2C* over-expression were associated with unfavourable survival outcomes [57]. Another study comparing patients with MM to controls identified 348 differentially expressed metabolism-related genes with prognostic features, of which most related to the proteasome, purine metabolism, cysteine metabolism, and methionine metabolism [58].

Whole exome sequencing on blood samples comparing patients resistant or refractory to both bortezomib and daratumumab were compared with those of patients responding to bortezomib treatment. Numerous significant variants were identified, including in the *PIK3CG* gene, which has metabolic cellular functions, including as a lipid kinase with altered effects influencing proliferation, angiogenesis, and drug resistance [59].

Multiple gene signatures, including those relating to hypoxia adaptation, protein folding, and mitochondrial respiration, were significantly altered in patients with MM, which was primary refractory to combination treatment including bortezomib compared to those with newly diagnosed MM [60]. These alterations may point towards further areas of significant metabolic change for targeted therapies. 

## 8. Conclusions and Future Perspectives

Cellular metabolism is complex and interlinked, and the metabolic alterations that occur in the oncogenesis of MM compared with the metabolic features seen in MGUS or in normal bone marrow remain only partially understood. Far less again is known about what further metabolic adaptations occur in the subsequent progression from newly diagnosed MM to relapsed/resistant MM, which help it to evade PIs and other previously effective treatments. Proteasome inhibition results in apoptosis of MM cells through a variety of mechanisms; and adaptations to evade these apoptotic processes and generate PI resistance have been shown to be influenced by many metabolic and metabolism-related factors, which can vary depending on the mechanism of action of the PI.

Whilst some individual metabolic processes and specific metabolic drug targets have been assessed in PI-sensitive and resistant MM (summarised in Table 1), this overall remains a relatively unexplored area with significant scope for further research. Investigating the metabolic alterations occurring in both the development of MM and in the later advance to PI resistance in depth will give us greater understanding of the pathogenesis of the disease and has the potential to identify adjuvant therapies to improve the efficacy of PIs in the management of resistant MM.

## Figures and Tables

**Figure 1 cancers-15-01682-f001:**
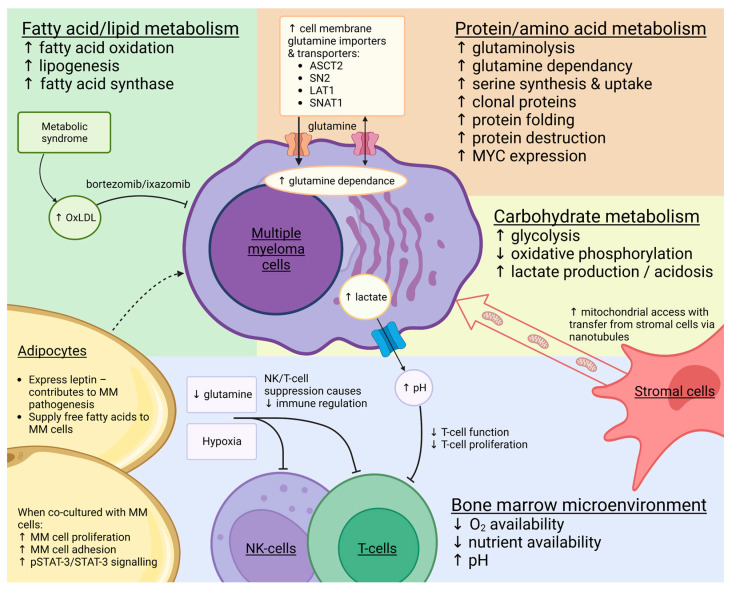
Summary of the main metabolic changes that develop in multiple myeloma (MM) and how systemic metabolic conditions and cells in the bone marrow microenvironment interact with and influence MM cell metabolism. ASCT2: alanine/serine/cysteine-preferring transporter 2; LAT1: L-type/large neutral amino acid transporter 1; OxLDL: oxidatively modified low-density lipoprotein; SN2: system N transporter; SNAT1: sodium-coupled neutral amino acid transporter 1.

**Figure 2 cancers-15-01682-f002:**
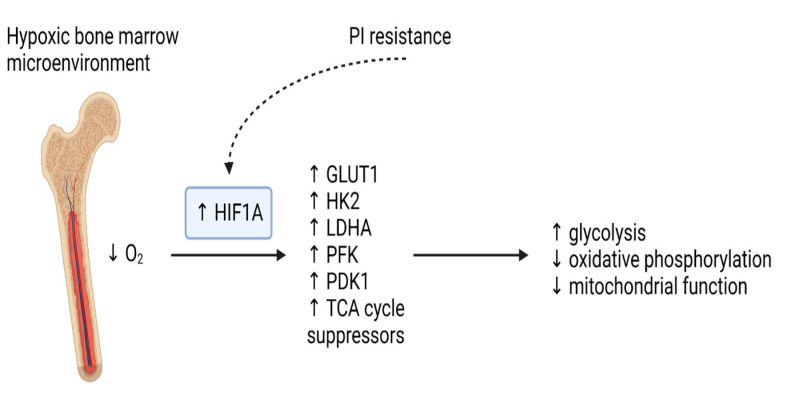
HIF1A expression is increased by both hypoxic conditions and the development of PI resistance, leading to upregulation of glucose transporters and glycolytic enzymes, which in turn promote a metabolic shift to increased glycolysis, along with suppression of oxidative phosphorylation and decreased mitochondrial function. GLUT1: glucose transporter 1; HIF1A: hypoxia-inducible factor 1 alpha; HK2: hexokinase 2; LDHA: lactate dehydrogenase A; PDK1: pyruvate dehydrogenase kinase-1; PFK: phosphofructokinase; PI: proteasome inhibitor; TCA: tricarboxylic acid.

**Table 1 cancers-15-01682-t001:** Inhibitors of MM cell metabolism.

Metabolic Pathway	Target	Compound	Reference
Glycolysis	PDK1	dichloroacetate	[12]
Glucose transport	GLUT4	Ritonavir	[19,21]
	Hexokinase 2	3-bromopyruvate/	
		2-deoxyglucose	[8,23]
	SGK1	GSK650394	[24]
Hypoxia	HIF1A	EZN-2968	[13,14]
Lactate transport	MCT1	α-cyano-4-hydroxycinnamic	[17]
Glutamine metabolism	glutaminase	Compound-968	[33]
	glutamine	L-asparaginase	[8]
	ASCT2	GPNA/CB-839	[34,35,36]
Serine synthesis	PGDH	NCT-503	[38]
Proline metabolism	PYCR1	paragline	[41]
Glutathione metabolism		Docosahexaenoic acid/ Eicosapentaenoic acid	[42]
Lipid metabolism	LDL	simvastatin	[47,48]
		orlistat/etomoxir	[17]
		Lovastatin/fenofibrate	[52]
	Sphingomyelin synthase	D609	[53]
		K145	[54]

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
