# Peer review of "Metabolic Alterations in Multiple Myeloma: From Oncogenesis to Proteasome Inhibitor Resistance"

_cancers, 2023, doi:10.3390/cancers15061682_

Round 1

Reviewer 1 Report

In this review article by Weir et al, the authors have presented a balanced overview of  the major metabolic changes that occur in MM pathogenesis and in the development of PI-resistance. Overall, the manuscript is quite well-written and would certainly be a good addition to the MM literature.

 Comments:

1.  Fig 1: Please include dendritic cells, T and NK cells and their contributions to metabolic changes in MM.

2. It is important to add the results from some of the recent trials (with NCT#s) related to the central theme of this manuscript. Please mention the limitations of such trials.

3. How are the combination therapies involving bortezomib affect the metabolic changes in MM?

4. It is recommended to mention the survival dependencies of some of the major metabolic enzymes discussed in the manuscript. May include survival analysis using the publicly available database.

5. How do  the metabolic changes described here affect the immune suppression in MM?

6. Section 4.3: Other amino acids: There were some clinical trials in the recent past targeting IDO1, the rate limiting enzyme in the Kyn pathway. If would be important to include a few lines about those trials indicating the outcome.

Reviewer 2 Report

In this review the authors focuses on the metabolic adaptations that occur in the development of monoclonal gammopathy of undetermined significance (MGUS), a precancerous stage that precedes MM, in the progression to symptomatic MM and in the subsequent development of PI resistance, and consider opportunities for therapeutic intervention.

In my opinion the review is well structured and well written. The weak part is related to the evolution from MGUS, SMM and MM which is slightly debated, and I think that also a dedicated figure (such as the Figure 1 ) would be an added value, to differentiate this preneoplastic state from the myeloma as treated e.g. by PI in terms of metabolic signatures/pathways.

Reviewer 3 Report

The manuscript of Wier et al is a very good review connecting the metabolic features of multiple myeloma cells and their resistance to treatment with proteasome inhibitors. The paper is well-written, the structure is correct, and it is easy to follow by the reader. The references are used properly. The figures and tables are useful and well-referenced in the text.

Author Response

We thank the reviewer for their comments